# Feature selection in functional data classification with recursive maxima hunting

**José L. Torrecilla**
Computer Science Department
Universidad Autónoma de Madrid
28049 Madrid, Spain
joseluis.torrecilla@uam.es

**Alberto Suárez**
Computer Science Department
Universidad Autónoma de Madrid
28049 Madrid, Spain
alberto.suarez@uam.es

## Abstract

Dimensionality reduction is one of the key issues in the design of effective machine learning methods for automatic induction. In this work, we introduce recursive maxima hunting (RMH) for variable selection in classification problems with functional data. In this context, variable selection techniques are especially attractive because they reduce the dimensionality, facilitate the interpretation and can improve the accuracy of the predictive models. The method, which is a recursive extension of maxima hunting (MH), performs variable selection by identifying the maxima of a relevance function, which measures the strength of the correlation of the predictor functional variable with the class label. At each stage, the information associated with the selected variable is removed by subtracting the conditional expectation of the process. The results of an extensive empirical evaluation are used to illustrate that, in the problems investigated, RMH has comparable or higher predictive accuracy than standard dimensionality reduction techniques, such as PCA and PLS, and state-of-the-art feature selection methods for functional data, such as maxima hunting.

## 1 Introduction

In many important prediction problems from different areas of application (medicine, environmental monitoring, etc.) the data are characterized by a function, instead of by a vector of attributes, as is commonly assumed in standard machine learning problems. Some examples of these types of data are functional magnetic resonance imaging (fMRI) (Grosenick et al., 2008) and near-infrared spectra (NIR) (Xiaobo et al., 2010). Therefore, it is important to develop methods for automatic induction that take into account the functional structure of the data (infinite dimension, high redundancy, etc.) (Ramsay and Silverman, 2005; Ferraty and Vieu, 2006). In this work, the problem of classification of functional data is addressed. For simplicity, we focus on binary classification problems (Baíllo et al., 2011). Nonetheless, the proposed method can be readily extended to a multiclass setting. Let $X(t), t \in [0, 1]$ be a continuous stochastic process in a probability space $(\Omega, \mathcal{F}, \mathbb{P})$. A functional datum $X_n(t)$ is a realization of this process (a trajectory). Let $\{X_n(t), Y_n\}_{n=1}^{N_{train}}, t \in [0, 1]$ be a set of trajectories labeled by the dichotomous variable $Y_n \in \{0, 1\}$. These trajectories come from one of two different populations; either $P_0$, when the label is $Y_n = 0$, or $P_1$, when the label is $Y_n = 1$. For instance, the data could be the ECG's from either healthy or sick persons ($P_0$ and $P_1$, respectively). The classification problem consist in deciding to which population a new unlabeled observation $X^{test}(t)$ belongs (e.g., to decide from his or her ECG whether a person is healthy or not). Specifically, we are interested in the problem of dimensionality reduction for functional data classification. The goal is to achieve the optimal discrimination performance using only a finite, small set of values from the trajectory as input to a standard classifier (in our work, $k$-nearest neighbors).

In general, to properly handle functional data, some kind of reduction of information is necessary. Standard dimensionality reduction methods in functional data analysis (FDA) are based on principal component analysis (PCA) (Ramsay and Silverman, 2005) or partial least squares (PLS) (Preda et al., 2007). In this work, we adopt a different approach based on variable selection (Guyon et al., 2006). The goal is to replace the complete function $X(t)$ by a $d$-dimensional vector $(X(t_1), \ldots, X(t_d))$ for a set of "suitable chosen" points $\{t_1, \ldots, t_d\}$ (for instance, instants in a heartbeat in ECG's), where $d$ is small.

Most previous work on feature selection in supervised learning with functional data is quite recent and focuses on regression problems; for instance, on the analysis of fMRI images (Grosenick et al., 2008; Ryali et al., 2010) and NIR spectra (Xiaobo et al., 2010). In particular, adaptations of *lasso* and other embedded methods have been proposed to this end (see, e.g., Kneip and Sarda (2011); Zhou et al. (2013); Aneiros and Vieu (2014)). In most cases, functional data are simply treated as high-dimensional vectors for which the standard methods apply. Specifically, Gómez-Verdejo et al. (2009) propose feature extraction from the functional trajectories before applying a multivariate variable selector based on measuring the mutual information. Similarly, Fernandez-Lozano et al. (2015) compare different standard feature selection techniques for image texture classification. The method of minimum Redundancy Maximum Relevance (mRMR) introduced by Ding and Peng (2005) has been applied to functional data in Berrendero et al. (2016a). In that work distance correlation (Székely et al., 2007) is used instead of mutual information to measure nonlinear dependencies, with good results. A fully functional perspective is adopted in Ferraty et al. (2010) and Delaigle et al. (2012). In these articles, a wrapper approach is used to select the optimal set of instants in which the trajectories should be monitored by minimizing a cross-validation estimate of the classification error. Berrendero et al. (2015) introduce a filter selection procedure based on computing the Mahalanobis distance and Reproducing Kernel Hilbert Space techniques. Logistic regression models have been applied to the problem of binary classification with functional data in Lindquist and McKeague (2009) and McKeague and Sen (2010), assuming Brownian and fractional Brownian trajectories, respectively. Finally, the selection of intervals or elementary functions instead of variables is addressed in Li and Yu (2008); Fraiman et al. (2016) or Tian and James (2013).

From the analysis of previous work one concludes that, in general, it is preferable, both in terms of accuracy and interpretability, to adopt a fully functional approach to the problem. In particular, if the data are characterized by functions that are continuous, values of the trajectory that are close to each other tend to be highly redundant and convey similar information. Therefore, if the value of the process at a particular instant has high discriminant capacity, one could think of discarding nearby values. This idea is exploited in maxima hunting (MH) (Berrendero et al., 2016b).

In this work, we introduce recursive Maxima Hunting (RMH), a novel variable selection method for feature selection in functional data classification that takes advantage of the good properties of MH while addressing some of its deficiencies. The extension of MH consists in removing the information conveyed by each selected local maximum before searching for the next one in a recursive manner. The rest of the paper is organized as follows: Maxima hunting for feature selection in classification problems with functional data is introduced in Section 2. Recursive maxima hunting, which is the method proposed in this work, is described in Section 3. The improvements that can be obtained with this novel feature selection method are analyzed in an exhaustive empirical evaluations whose results are presented and discussed in Section 4.

## 2  Maxima Hunting

Maxima hunting (MH) is a method for feature selection in functional classification based on measuring dependencies between values selected from $\{X(t), t \in [0, 1]\}$ and the response variable (Berrendero et al., 2016b). In particular, one selects the values $\{X(t_1), \ldots, X(t_d)\}$ whose dependence with the class label (i.e., the response variable) is locally maximal. Different measures of dependency can be used for this purpose. In Berrendero et al. (2016b), the authors propose the distance correlation (Székely et al., 2007). The distance covariance between the random variables $X \in \mathbb{R}^p$ and $Y \in \mathbb{R}^q$, whose components are assumed to have finite first-order moments, is

$$\mathcal{V}^2(X, Y) = \int_{\mathbb{R}^{p+q}} \mid \varphi_{X,Y}(u, v) - \varphi_X(u)\varphi_Y(v) \mid^2 w(u, v)dudv, \tag{1}$$

where $\varphi_{X,Y}, \varphi_X, \varphi_Y$ are the characteristic functions of $(X, Y)$, $X$ and $Y$, respectively, $w(u, v) = (c_p c_q |u|_p^{1+p} |v|_q^{1+q})^{-1}$, $c_d = \frac{\pi^{(1+d)/2}}{\Gamma((1+d)/2)}$ is half the surface area of the unit sphere in $\mathbb{R}^{d+1}$, and $|\cdot|_d$ stands for the Euclidean norm in $\mathbb{R}^d$.

In terms of $\mathcal{V}^2(X, Y)$, the square of the distance correlation is

$$\mathcal{R}^2(X, Y) = \begin{cases} \frac{\mathcal{V}^2(X,Y)}{\sqrt{\mathcal{V}^2(X,X)\mathcal{V}^2(Y,Y)}}, & \mathcal{V}^2(X)\mathcal{V}^2(Y) > 0 \\ 0, & \mathcal{V}^2(X)\mathcal{V}^2(Y) = 0. \end{cases} \tag{2}$$

The distance correlation is a measure of statistical independence; that is, $\mathcal{R}^2(X, Y) = 0$ if and only if $X$ and $Y$ are independent. Besides being defined for random variables of different dimensions, it has other valuable properties. In particular, it is rotationally invariant and scale equivariant (Székely and Rizzo, 2012). A further advantage over other measures of independence, such as the mutual information, is that the distance correlation can be readily estimated using a plug-in estimator that does not involve any parameter tuning. The almost sure convergence of the estimator $\mathcal{V}_n^2$ is proved in Székely et al. (2007, Thm. 2).

To summarize, in maxima hunting, one selects the $d$ different local maxima of the distance correlation between $X(t)$, the values of random process at different instants $t \in [0, 1]$, and the response variable

$$X(t_i) = \underset{t \in [0,1]}{\operatorname{argmax}} \mathcal{R}^2(X(t), Y), \quad i = 1, 2, \dots, d. \tag{3}$$

Maxima Hunting is easy to interpret. It is also well-motivated from the point of view of FDA, because it takes advantage of functional properties of the data, such as continuity, which implies that similar information is conveyed by the values of the function at neighboring points. In spite of the simplicity of the method, it naturally accounts for the relevance and redundancy trade-off in feature selection (Yu and Liu, 2004): the local maxima (3) are relevant for discrimination. Points around them, which do not maximize the distance correlation with the class label, are automatically excluded. Furthermore, it is also possible to derive a uniform convergence result, which provides additional theoretical support for the method. Finally, the empirical investigation carried out in Berrendero et al. (2016b) shows that MH performs well in standard benchmark classification problems for functional data. In fact, for some problems, one can show that the optimal (Bayes) classification rules depends only on the maxima of $\mathcal{R}^2(X(t), Y)$.

However, maxima hunting presents also some limitations. First, it is not always a simple task to estimate the local maxima, especially in functions that are very smooth or that vary abruptly. Furthermore, there is no guarantee that different maxima are not redundant. In most cases, the local maxima of $\mathcal{R}^2(X(t), Y)$ are indeed relevant for classification. However, there are important points for which this quantity does not attain a maximum.

As an example, consider the family of classification problems introduced in Berrendero et al. (2016b, Prop. 3), in which the goal is to discriminate trajectories generated by a standard Brownian motion process, $B(t)$, and trajectories from the process $B(t) + \Phi_{m,k}(t)$, where

$$\Phi_{m,k}(t) = \int_0^t \sqrt{2^{m-1}} \left[ \mathbb{I}_{\left(\frac{2k-2}{2^m}, \frac{2k-1}{2^m}\right)}(s) - \mathbb{I}_{\left(\frac{2k-1}{2^m}, \frac{2k}{2^m}\right)}(s) \right] ds, \quad m, k \in \mathbb{N}, 1 \le k \le 2^{m-1}. \tag{4}$$

Assuming a balanced class distribution ($\mathbb{P}(Y = 0) = \mathbb{P}(Y = 1) = 1/2$), the optimal classification rule is $g^*(x) = 1$ if and only if $\left( X\left(\frac{2k-1}{2^m}\right) - X\left(\frac{2k-2}{2^m}\right) \right) + \left( X\left(\frac{2k-1}{2^m}\right) - X\left(\frac{2k}{2^m}\right) \right) > \frac{1}{\sqrt{2^{m+1}}}$.

The optimal classification error is $L^* = 1 - \text{normcdf}\left( \frac{\|\Phi'_{m,k}(t)\|}{2} \right) = 1 - \text{normcdf}\left(\frac{1}{2}\right) \simeq 0.3085$,

where, $\|\cdot\|$ denotes the $L^2[0, 1]$ norm, and $\text{normcdf}(\cdot)$ is the cumulative distribution function of the standard normal. The relevance function has a single maximum at $X\left(\frac{2k-1}{2^m}\right)$. However, the Bayes classification rule involves three relevant variables, two of which are clearly not maxima of $\mathcal{R}^2(X(t), Y)$. In spite of the simplicity of these types of functional classification problems, they are important to analyze, because the set of functions $\Phi_{m,k}$, with $m > 0$ and $k > 0$ form an orthonormal basis of the Dirichlet space $\mathcal{D}[0, 1]$, the space of continuous functions whose derivatives are in $L^2[0, 1]$. Furthermore, this space is the reproducing kernel Hilbert space associated with Brownian motion and plays and important role in functional classification (Mörters and Peres, 2010; Berrendero et al., 2015). In fact, any trend in the Brownian process can be approximated by a linear combination or by a mixture of $\Phi_{m,k}(t)$.

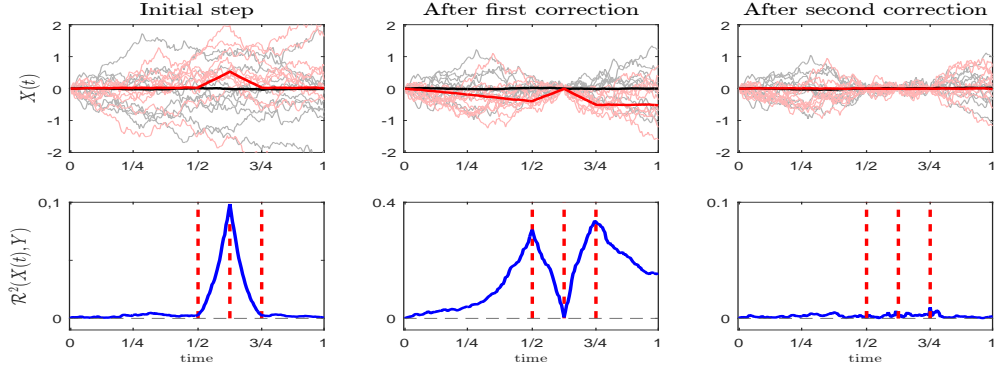

Figure 1: *First row*: Individual and average trajectories for the classification of $B(t)$ vs. $B(t) + 2\Phi_{3,3}(t)$ initially (left) and after the first (center) and second (right) corrections. *Second row*: Values of $\mathcal{R}^2(X(t), Y)$ as a function of t. The variables required for optimal classification are marked with vertical dashed lines.

To illustrate the workings of maxima hunting and its limitations we analyze in detail the classification problem $B(t)$ vs. $B(t) + 2\Phi_{3,3}(t)$, which is of the type considered above. In this case, the optimal classification rule depends on the maximum $X(5/8)$, and on $X(1/2)$ and $X(3/4)$, which are not maxima, and would therefore not be selected by the MH algorithm. The optimal error is $L^* = 15.87\%$. To illustrate the importance of selecting all the relevant variables, we perform simulations in which we compare the accuracy of the linear Fisher discriminant with the maxima hunting selection, and with the optimal variable selection procedures. In these experiments, independent training and test samples of size 1000 are generated. The values reported are averages over 100 independent runs. Standard deviations are given between parentheses. The average prediction error when only the maximum of the trajectories is considered is $37.63\%(1.44\%)$. When all three variables are used the empirical error is $15.98\%(1\%)$, which is close to the Bayes error. When other points in addition to the maximum are used (i.e., $(X(t_1), X(5/8), X(t_2)$, with $t_1$ and $t_2$ randomly chosen so that $0 \leq t_1 < 5/8 < t_2 \leq 1$) the average classification error is $22.32\%(2.18\%)$. In the top leftmost plot of Figure 1 trajectories from both classes, together with the corresponding averages (thick lines) are shown. The relevance function $\mathcal{R}^2(X(t), Y)$ is plotted below. The relevant variables, which are required for optimal classification, are marked by dashed vertical lines.

## 3   Recursive Maxima Hunting

As a variable selection process, MH avoids, at least partially, the redundancy introduced by the continuity of the functions that characterize the instances. However, this local approach cannot detect redundancies among different local maxima. Furthermore, there could be points in the trajectory that do not correspond to maxima of the relevance function, but which are relevant when considered jointly with the maxima. The goal of recursive maxima hunting (RMH) is to select the maxima of $\mathcal{R}^2(X(t), Y)$ in a recursive manner by removing at each step the information associated to the most recently selected maximum. This avoids the influence of previously selected maxima, which can obscure ulterior dependencies. The influence of a selected variable $X(t_0)$ on the rest of the trajectory can be eliminated by subtracting the conditional expectation $\mathbb{E}(X(t)|X(t_0))$ from $X(t)$. Assuming that the underlying process is Brownian

$$\mathbb{E}(X(t)|X(t_0)) = \frac{\min(t, t_0)}{t_0} X(t_0), \quad t \in [0, 1]. \tag{5}$$

In the subsequent iterations, there are two intervals: $[t, t_0]$ and $[t_0, 1]$. Conditioned on the value at $X(t_0)$, the process in the interval $[t_0, 1]$ is still Brownian motion. By contrast, for the interval $[0, t_0]$ the process is a Brownian bridge, whose conditional expectation is

$$\mathbb{E}(X(t)|X(t_0) = \frac{\min(t, t_0) - t\,t_0}{t_0(1 - t_0)} X(t_0) = \begin{cases} \frac{t}{t_0} X(t_0), & t < t_0 \\ \frac{1-t}{1-t_0} X(t_0), & t > t_0. \end{cases} \tag{6}$$

As illustrated by the results in the experimental section, the Brownian hypothesis is a robust assumption. Nevertheless, if additional information on the underlying stochastic processes is available, it can

be incorporated to the algorithm during the calculation of the conditional expectation in Equations (5) and (6).

The center and right plots in Figure 1 illustrate the behavior of RMH in the example described in the previous section. The top center plot displays the trajectories and corresponding averages (thick lines) for both classes after applying the correction (5) with $t_0 = 5/8$, which is the first maximum of the distance correlation function (bottom leftmost plot in Figure 1). The variable $X(5/8)$ is clearly uninformative once this correction has been applied. The distance correlation $\mathcal{R}^2(X(t), Y)$ for the corrected trajectories is displayed in the bottom center plot. Also in this plot the relevant variables are marked by vertical dashed lines. It is clear that the subsequent local maxima at $t = 1/2$, in the subinterval $[0, 5/8]$, and at $t = 3/4$, in the subinterval, and $[5/8, 1]$ correspond to the remaining relevant variables. The last column shows the corresponding plots after the correction is applied anew (equations (6) with $t_0 = 1/2$ in $[0, 5/8]$ and (5) with $t_0 = 3/4$ in $[5/8, 1]$). After this second correction, the discriminant information has been removed. In consequence, the distance correlation function, up to sample fluctuations, is zero.

An important issue in the application of this method is how to decide when to stop the recursive search. The goal is to avoid including irrelevant and/or redundant variables. To address the first problem, we only include maxima that are sufficiently prominent $\mathcal{R}^2(X(t_{max}), Y) > s$, where $0 < s < 1$ can be used to gauge the relative importance of the maximum. Redundancy is avoided by excluding points around a selected maximum $t_{max}$ for which $\mathcal{R}^2(X(t_{max}), X(t)) \geq r$, for some redundancy threshold $0 < r < 1$, which is typically close to one. As a result of these two conditions only a finite (typically small) number of variables are selected. This data-driven stopping criterion avoids the need to set the number of selected variables beforehand or to determine this number by a costly validation procedure. The sensitivity of the results to the values of $r$ and $s$ will be studied in Section 4. Nonetheless, RMH has a good and robust performance for a wide range of reasonable values of these parameters ($r$ close to 1 and $s$ close to 0). The pseudocode of the RMH algorithm is given in Algorithm 1.

---

**Algorithm 1** Recursive Maxima Hunting

1: **function RMH**$(X(t), Y)$
2:     $\mathbf{t}^* \leftarrow [\,]$                                   $\triangleright$ Vector of selected points initially empty
3:     RMH_rec$(X(t), Y, 0, 1)$               $\triangleright$ Recursive search of the maxima of $\mathcal{R}^2(X(t), Y)$
4:     **return $\mathbf{t}^*$**                                $\triangleright$ Vector of selected points
5: **end function**
6: **procedure RMH_REC**$(X(t), Y, t_{inf}, t_{sup})$
7:     $t_{max} \leftarrow \underset{t_{inf} \leq t \leq t_{sup}}{\mathrm{argmax}} \; \{ \mathcal{R}^2(X(t), Y) \}$
8:     **if** $\mathcal{R}^2(X(t_{max}), Y) > s$ **then**
9:         $\mathbf{t}^* \leftarrow [\mathbf{t}^* \; t_{max}]$              $\triangleright$ Include $t_{max}$ in $\mathbf{t}^*$ the vector of selected points
10:         $X(t) \leftarrow X(t) - \mathbb{E}(X(t) \mid X(t_{max})), \; t \in [t_{inf}, t_{sup}]$  $\triangleright$ Correction of type (5) or (6) as required
11:     **else**
12:         **return**
13:     **end if**
14:     $\triangleright$ Exclude redundant points to the left of $t_{max}$
15:     $t_{max}^- \leftarrow \underset{t_{inf} \leq t < t_{max}}{\max} \{ t : \mathcal{R}^2(X(t_{max}), X(t)) \leq r \}$
16:     **if** $t_{max}^- > t_{inf}$ **then**
17:         RMH_rec$(X(t), Y, t_{inf}, t_{max}^-)$                  $\triangleright$ Recursion on left subinterval
18:     **end if**
19:     $\triangleright$ Exclude redundant points to the right of $t_{max}$
20:     $t_{max}^+ \leftarrow \underset{t_{max} < t \leq t_{sup}}{\min} \{ t : \mathcal{R}^2(X(t_{max}), X(t)) \leq r \}$
21:     **if** $t_{max}^+ < t_{sup}$ **then**
22:         RMH_rec$(X(t), Y, t_{max}^+, t_{sup})$                 $\triangleright$ Recursion on right subinterval
23:     **end if**
24:     **return**
25: **end procedure**

---

## 4 Empirical study

To assess the performance of RMH, we have carried out experiments in simulated and real-world data in which it is compared with some well-established dimensionality reduction methods, such as PCA (Ramsay and Silverman, 2005) and partial least squares (Delaigle and Hall, 2012b), and with Maxima Hunting (Berrendero et al., 2016b). In these experiments, $k$-nearest neighbors (kNN) with the Euclidean distance is used for classification. kNN has been selected because it is a simple, nonparametric classifier with reasonable overall predictive accuracy. The value $k$ in kNN is selected by 10-fold CV from integers in $[1, \sqrt{N_{train}}]$, where $N_{train}$ is the size of the training set. Since RMH is a filter method for variable selection, the results are expected to be similar when other types of classifiers are used. As a reference, the results of kNN using complete trajectories (i.e., without dimensionality reduction) are also reported. This approach is referred to as *Base*. Note that, in this case, the performance of kNN need not be optimal because of the presence of irrelevant attributes.

RMH requires determining the values of two hyperparameters: the redundancy threshold $r$ ($0 < r < 1$ typically close to 1), and the relevance threshold $s$ ($0 < s < 1$ typically close to 0). Through extensive simulations we have observed that RMH is quite robust for a wide range of appropriate values of these parameters. In particular, the results are very similar for values of r in the interval $[0.75, 0.95]$. The predictive accuracy is somewhat more sensitive to the choice of $s$: If the value of $s$ is too small, irrelevant variables can be selected. If $s$ is too large, it is possible that relevant points are excluded. For most of the experiments performed, the optimal values of $s$ are between 0.025 and 0.1. In view of these observations, the experiments are made using $r = 0.8$. The value of $s$ is selected from the set $\{0.025, 0.05, 0.1\}$ by 10-fold CV. A more careful determination of $r$ and $s$ is beneficial, especially in some extreme problems (e.g., with very smooth or with rapidly-varying trajectories). In RMH, the number of selected variables, which is not determined beforehand, depends indirectly on the values of $r$ and $s$. In the other methods, the number of selected variables is determined using 10-fold CV, with maximum of 30.

A first batch of experiments is carried out on simulated data generated from the model

$$\begin{cases} P_0 : B(t) & , \quad t \in [0,1] \\ P_1 : B(t) + m(t) & , \quad t \in [0,1] \end{cases} ,$$

where $B(t)$ is standard Brownian motion, $m(t)$ is a deterministic trend, and $\mathbb{P}(Y = 0) = \mathbb{P}(Y = 1) = 1/2$. Using Berrendero et al. (2015, Theorem 2), it is possible to compute the optimal classification rules $g^*$ and the corresponding Bayes errors $L^*$. To ensure a wide coverage, we consider two problems in which the Bayes rule depends only on a few variables and two problems in which complete trajectories are needed for optimal classification: (i) *Peak*: $m(t) = 2\Phi_{3,3}(t)$. The optimal rule depends only on $X(1/2)$, $X(5/8)$ and $X(3/4)$. The Bayes error is $L^* \simeq 0.1587$. This is the example analyzed in the previous section. (ii) *Peak2*: $m(t) = 2\Phi_{3,2}(t) + 3\Phi_{3,3}(t) - 2\Phi_{2,2}(t)$. The optimal rule depends only on $X(1/4), X(3/8), X(1/2), X(5/8), X(3/4)$, and $X(1)$. The Bayes error is $L^* \simeq 0.0196$. (iii) *Square*: $m(t) = 2t^2$. The Bayes error is $L^* \simeq 0.1241$. (iv) *Sin*: $m(t) = 1/2\sin(2\phi t)$. The Bayes error is $L^* \simeq 0.1333$. In Figure 2 we have plotted some trajectories corresponding to class 1 instances, together with their corresponding averages (thick lines). Class 0 trajectories are realizations of a standard Brownian process. In these experiments, training samples of different sizes ($N_{train} = \{50, 100, 200, 500, 1000\}$) and an independent test set of size 1000 are generated. The trajectories are discretized in 200 points. Half of the trajectories belong to each class in both the training and test sets. The values reported are averages over 200 independent repetitions.

Figure 3 displays the average classification error (first row) and the average number of selected variable /components (second row) as a function of the training sample size for each model and classification method. Horizontal dashed lines are used to indicate the Bayes error level in the different problems. From the results reported in Figure 3, one concludes that RMH has the best overall performance. It is always more accurate than the *Base* method. This observation justifies performing variable selection not only for the sake of dimensionality reduction, but also to improve the classification accuracy. RMH is also better than the original MH in all the problems investigated: there is both an improvement of the prediction error, and a reduction of the numeber of variables used for classification. In *peak* and *peak2*, problems in which the relevant variables are known, RMH generally selects the correct ones. As expected, PLS performs better than PCA. However, both MH and RMH outperform these projection methods, except in *sin*, where their accuracies are similar. Both PLS and RMH are effective dimensionality reduction methods with comparable performance.

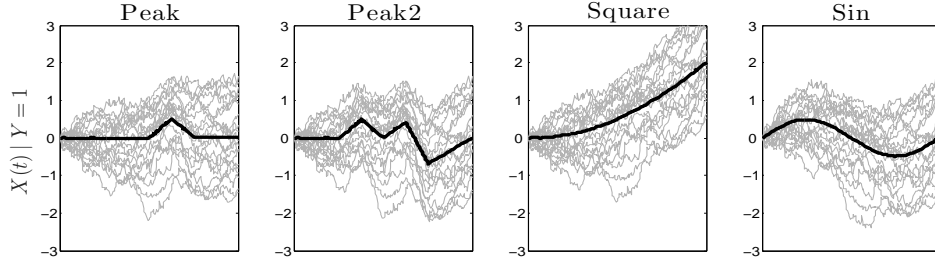

Figure 2: Class 1 trajectories and averages (thick lines) for the different synthetic problems.

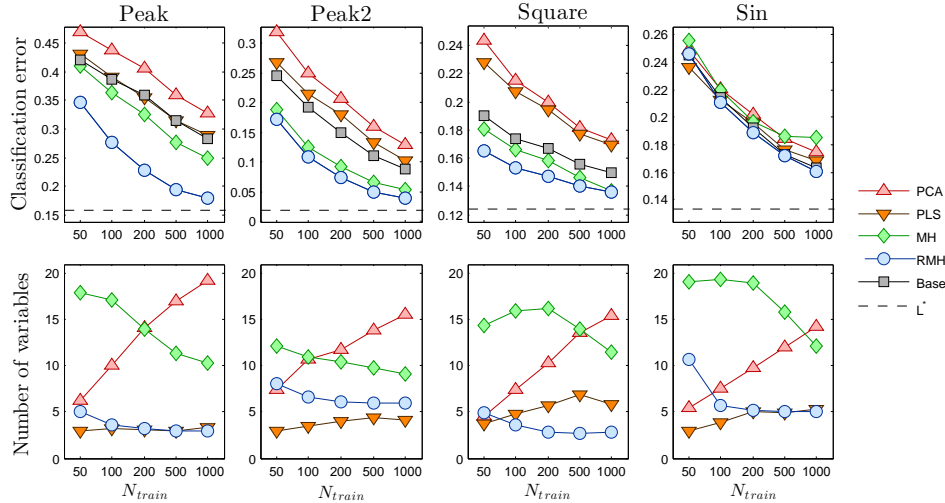

Figure 3: Average classification error (first row) and average number of selected variables/components (second row) as a function of the size of the training.

However, the components selected in PLS are, in general, more difficult to interpret because they involve whole trajectories. Finally, the accuracy of RMH is very close to the Bayes level for higher sample sizes, even when the optimal rule requires using complete trajectories (*square* and *sin*).

To assess the performance of RMH in real-world functional classification problems, we have carried out a second batch of experiments in four datasets, which are commonly used as benchmarks in the FDA literature. Instances in *Growth* correspond to curves of the heights of 54 girls and 38 boys from the *Berkeley Growth Study*. Observations are discretized in 31 non-equidistant ages between 1 and 18 years (Ramsay and Silverman, 2005; Mosler and Mozharovskyi, 2014). The *Tecator* dataset consists of 215 near-infrared absorbance spectra of finely chopped meat. The spectral curves consist of 100 equally spaced points. The class labels are determined in terms of fat content (above or below 20%). The curves are fairly smooth. In consequence, we have followed the general recommendation and used the second derivative for classification (Ferraty and Vieu, 2006; Galeano et al., 2014). The *Phoneme* data consists of 4509 log-periodograms observed at 256 equidistant points. Here, we consider the binary problem of distinguishing between the phonemes "aa" (695) and "ao" (1022) (Galeano et al., 2014). Following Delaigle and Hall (2012a), the curves are smoothed with a local linear method and truncated to the first 50 variables. The *Medflies* are records of daily egg-laying patterns of a thousand flies. The goal is to discriminate between short- and long-lived flies. Following Mosler and Mozharovskyi (2014), curves equal to zero are excluded. There are 512 30-day curves (starting from day 5) of flies who live at most 34 days, 266 of these are long-lived (reach the day 44). The classes in *Growth* and *Tecator* are well separated. In consequence, they are relatively easy problems. By contrast, *Phoneme* and *Medflies* are notoriously difficult classification tasks. Some trajectories of each problem and each class, together with the corresponding averages (thick lines), are plotted in Figure 4. To estimate the classification error, the datasets are partitioned at random into a training set (with $2/3$ of the observations) and a test set ($1/3$). This procedure is repeated 200 times. The boxplots of the results for each dataset and method are shown in Figure 5. Errors are shown in first row and the number of selected variables/components in the second one. From

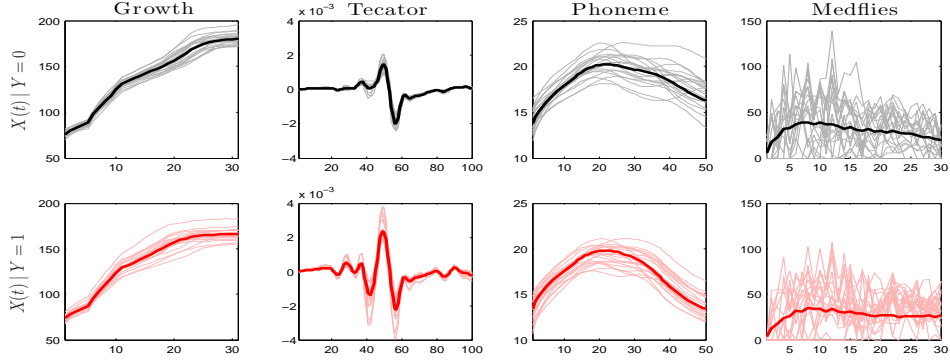

Figure 4: Trajectories for each of the classes and their corresponding averages (thick lines).

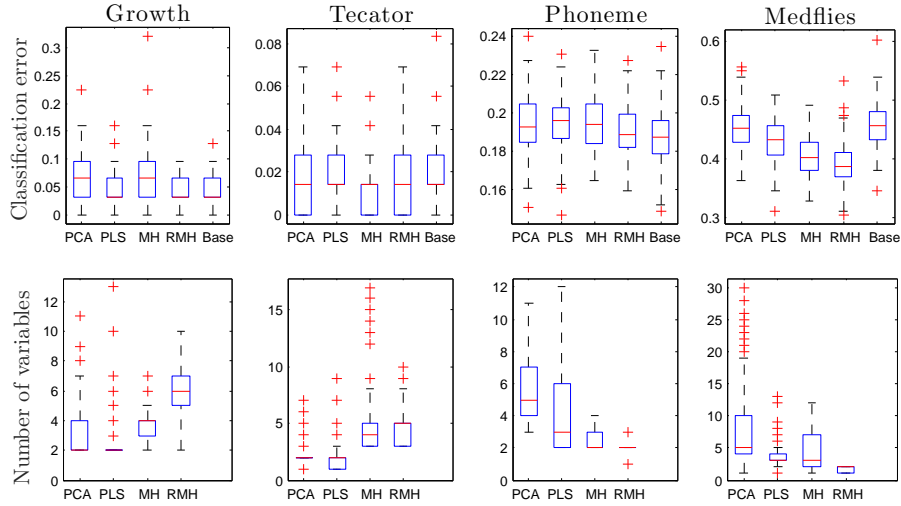

Figure 5: Classification error (first row) and number of variables/components selected (second row) by RMH.

these results we observe that, in general, dimensionality reduction is effective: the accuracy of the four considered methods is similar or better than the *Base* method, in which complete trajectories are used for classification. In particular, *Base* does not perform well when the trajectories are not smooth (*Medflies*). The best overall performance corresponds to RMH. In the easy problems (*Growth* and *Tecator*), all methods behave similarly and give good results. In *Growth*, RMH is slightly more accurate. However, it tends to select more variables than the other methods. In the more difficult problems, (*Phoneme* and *Medflies*), RMH yields very accurate predictions while selecting only two variables. In these problems it exhibits the best performance, except in *Phoneme*, where *Base* is more accurate. The variables selected by RMH and MH are directly interpretable, which is an advantage over projection-based methods (PCA, PLS). Finally, let us point out that the accuracy of RMH is comparable and often better that state-of-the-art functional classification methods. See, for instance, Berrendero et al. (2016a); Delaigle et al. (2012); Delaigle and Hall (2012a); Mosler and Mozharovskyi (2014); Galeano et al. (2014). In most of these works no dimensionality reduction is applied. Nevertheless, these comparisons must be done carefully because the evaluation protocol and the classifiers used vary in the different studies. In any case, RMH is a filter method, which means that it could be more effective if used in combination with other types of classifiers or adapted and used as a wrapper or, even, as an embedded variable selection method.

## Acknowledgments

The authors thank Dr. José R. Berrendero for his insightful suggestions. We also acknowledge financial support from the Spanish Ministry of Economy and Competitiveness, project TIN2013-42351-P and from the Regional Government of Madrid, CASI-CAM-CM project (S2013/ICE-2845).

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
