[Reviews · NeurIPS 2016]

Reviewer 1

Summary

The paper describes a novel algorithm they call recursive maxima hunting (RMH) to perform variable selection in time-series data for classification. They demonstrate the algorithm on a variety of standard examples and compare it with standard dim red techniques including PCA and PLS.

Qualitative Assessment

The RMH method is an interesting one and the paper seems well executed. Certainly, functional data has dependencies that are distinct from a purely statistic description. However, the RMH method has many limitations and may not be widely useful -- it requires that functional data is perfectly aligned in time (i.e., we know precisely when t = 0 occurs). For a stronger paper, I would liked to have seem a more comprehensive analysis of the method with respect to various noise models, as well as to temporal jitter in the time of the measurements (variables). Also, perhaps one demo with a non-synthetic dataset (like the ones mentioned in the intro, perhaps?) would be really good to see.

Confidence in this Review

3-Expert (read the paper in detail, know the area, quite certain of my opinion)


Reviewer 2

Summary

The authors build on a method called Maxima Hunting for functional feature selection by proposing a recursive Maxima Hunting method that makes it possible to catch time points that would be ignored otherwise.

Qualitative Assessment

First let me say that while I know feature selection well, I haven't had the opportunity to read a lot of papers about functional feature selection. Therefore my background here is limited in terms of references. The paper is well organized and clear, in my opinion even though ironically wrt the contents it has a few redundancies - but that's fine. The experiments seem to be carried out thoroughly with one exception as the authors note themselves: the parameters s and r seem to be chosen a little arbitrarily and in my opinion more values of those should have been included in the cross validation procedure. It is also not very clear whether these values are fixed or not during the experiments on real data. That would need to be explained better. Overall the method seems elegant to me and I was quite convinced by the motivating example that uses the brownian bridge throughout the paper. From this example it seems clear that MH can indeed be improved upon. I also appreciated the fact that the authors compared their method with quite a few competitors although I was surprised to see that methods involving wavelets were not including (again, not an expert in the field). Whether the method really outperforms competition is not very clear from the experiments and maybe that would need a bit more analysis on the authors' part. However: 1. results on (what seems to be well) simulated data are convincing; 2. on "medflies" and "phoneme" I find it interesting that not only the classification results are better but that they are so using a small number of covariates - that, as the authors note, are interpretable as opposed to new features created by, e.g. PCA. I can imagine that having a small number of time points to look at might be important for the application. For these reasons, I'm leaning towards accepting this paper. I have a few minor comments and minutiae that the authors might want to look at: - in the algorithm, where does theta come from? - in the algorithm, how are t_inf and t_sup chosen? 158: do you mean [0, t_0]? 205: these results 254: remove “an” 258: where “it”? 259: “as opposed to”? (a bunch of other minutiae in the last paragraph)

Confidence in this Review

2-Confident (read it all; understood it all reasonably well)


Reviewer 3

Summary

This paper presents an approach, called Recursive Maxima Hunting (RMH), for selecting features for functional data classification. The proposed method is shown to yield good results on simulated and real data. Although it is an adaptation of an existing method, this work remains very interesting.

Qualitative Assessment

This paper presents an approach, called Recursive Maxima Hunting (RMH), for selecting features for functional data classification. RMH is an adaptation of the Maxima Hunting (MH) approach, which selects features having a relevance score that is locally maximum (the distance correlation being used as relevance score). One drawback of the MH approach is that it misses the features that are not individually relevant (and are thus not local maxima) but that become relevant when combined with other relevant features. To address this problem, RMH selects the features in a recursive way, and at each stage the data are corrected by removing the information brought by the last selected feature. In practice, the method assumes that the underlying process is a Brownian motion and corrects the data by subtracting the conditional expectation of the process given the value of the last selected feature. The proposed approach was applied on 4 simulated datasets and 4 real datasets, and yields good results. Although it is an adaptation of an existing method, this is a very interesting work. Overall, the paper is well structured and clearly written. Some comments: - I am not sure to understand why one needs to exclude redundant points around a selected maximum (lines 15 and 20 of Algorithm 1). Isn’t the redundancy problem addressed by doing the data correction? For example, in Figure 2, one can see that the points in the close vicinity of t=5/8 have a very low distance correlation. It thus seems that one can hardly find redundant points after the correction. If it is not always the case, the authors should show an example that would help to understand why removing neighbouring points is needed. - The conditional expectation of the Brownian motion is t/t0 * X(t0) while Equation (7) says it is min(t,t0)/t0 * X(t0). If it is not a typo, the authors should discuss why they chose that correction. - It would be interesting to know the sample sizes of the real datasets. Typos / others - Line 30: consistis —> consists - Line 72: searching from the next one —> searching for - Equation (2): two different notations are used for the distance variance. - Line 91: indepentence —> independence - Line 181: reasoble —> reasonable - Line 182: RMH algorithms —> RMH algorithm - Lines 15 and 20 of Algorithm 1: what is theta? - Lines 208-209: this sentence is not clear, since in RMH the number of selected features actually depends on some stopping criterion (i.e. the values of r and s). - Figure 4: on most x-axis labels, 50 is written instead of 1000 (for the sample size).

Confidence in this Review

1-Less confident (might not have understood significant parts)


Reviewer 4

Summary

This paper provide a nice extension of Maxima Hunting technique for variable selection and compare it with standard technique on a well suited empirical study as well as on real data.

Qualitative Assessment

The paper is clear and very easy to read, the work is well-placed in the literature and the results well-connected to existing ones. Unfortunately, the method did not demonstrate dramatic improvement in terms of performances compare to standard ones.. Moreover, I would appreciate if the authors could discuss a bit more the brownianity assumption of the underlying process, which could be restrictive.

Confidence in this Review

3-Expert (read the paper in detail, know the area, quite certain of my opinion)


Reviewer 5

Summary

This paper improves variable selection for functional data classification by extending Maxima Hunting to Recursive Maxima Hunting. Its contribution lies in assuming the process is Brownian, allowing the effect of each selected variable to be removed from the relevance function prior to selecting the next variable. This promotes "independence" among the selected variables, which is clearly desirable in dimensionality reduction problems.

Qualitative Assessment

The idea of using conditional expectation to improve MH into RMH is excellent. It comes at the cost of assuming Brownian motion functional data, but improved real data performance still seems to be obtained. Presentation issues: 1. Equation (3) is not a definition of a set of local maxima - it defines the set of global maxima. Please use a more rigorous definition. 2. Lines 158-159: Needs to be rewritten due to grammatical errors. Possible rewrite: "In the subsequent iterations, there are two intervals: [t,t0] and [t0,1]. Conditioned on the value at X(t0), the process in the interval [t0,1] is still Brownian motion." 3. Section 4.2 needs to be proofread, as it contains a large number of grammatical/spelling errors and examples of unclear/clunky phrasing.

Confidence in this Review

3-Expert (read the paper in detail, know the area, quite certain of my opinion)


Reviewer 6

Summary

The paper proposes a novel method of variable selection for functional classification. The authors argue cogently for a fully functional approach (as opposed to treating functional data as high-dimensional vectors), and build off of the theoretically well-motivated (and easily intepreted) technique of maxima hunting (MH). Their proposal is to recursively apply MH, removing in some sense the information carried by the maximum identified in each stage of the recursive process. They then demonstrate on a salient class of learning problems that their method of recursive maxima hunting (RMH) dominates other classifiers.

Qualitative Assessment

The paper is very well-written and closely argued. The method proposed is well-motivated and on the face of it quite promising. Technical quality The method is well-motivated. The only obvious objection might have been to the tunable parameters r and s corresponding to a redundancy and relevance threshold. However, the authors argue for plausible a priori bounds on their potential values, and suggest sensible procedures for tuning the hyperparameters. Novelty/originality To the best of my knowledge, no one has previously suggested using MH in this way. The method is new. Potential impact Given its performance, the easy interpretation of results, and the substantial theoretical virtues that recommend it, this is a method likely to be broadly adopted. Clarity and presentation The paper was a pleasure to read. I have only one minor suggestion. Though it is certainly clear from Algorithm 1, it might be helpful at the start of Section 3 to explicitly indicate that the removal of the information associated with the most recently selected maximum involves simply subtracting off the expectation value given in (7) or (8).

Confidence in this Review

2-Confident (read it all; understood it all reasonably well)